# Double Heterozygous Pathogenic Variants in *TP53* and *CHEK2* in Boy with Undifferentiated Embryonal Sarcoma of the Liver

**DOI:** 10.3390/ijms252111489

**Published:** 2024-10-25

**Authors:** Michaela Kuhlen, Tina Schaller, Sebastian Dintner, Nicole Stadler, Thomas G. Hofmann, Maximilian Schmutz, Rainer Claus, Michael C. Frühwald, Monika M. Golas

**Affiliations:** 1Pediatrics and Adolescent Medicine, Faculty of Medicine, University of Augsburg, 86156 Augsburg, Germany; 2Pathology, Faculty of Medicine, University of Augsburg, 86156 Augsburg, Germany; 3Institute of Toxicology, University Medical Center of the Johannes Gutenberg University Mainz, 55131 Mainz, Germany; 4Hematology and Oncology, Faculty of Medicine, University of Augsburg, 86156 Augsburg, Germany; 5Comprehensive Cancer Center Augsburg, Faculty of Medicine, University of Augsburg, 86156 Augsburg, Germany; 6Human Genetics, Faculty of Medicine, University of Augsburg, 86156 Augsburg, Germany

**Keywords:** *TP53*, *CHEK2*, double heterozygosity, embryonal sarcoma of the liver, synthetic lethality

## Abstract

Undifferentiated embryonal sarcoma of the liver is a rare mesenchymal malignancy that predominantly occurs in children. The relationship between this tumor entity and germline pathogenic variants (PVs) remains undefined. Here, we present the clinical case of a male patient diagnosed with undifferentiated embryonal sarcoma of the liver. Both germline and tumor samples were analyzed using next-generation sequencing. In the tumor tissue, PVs in *TP53* (NM_000546.5):c.532del p.(His178Thrfs*69) and *CHEK2* (NM_007194.4):c.85C>T p.(Gln29*) were identified, with both confirmed to be of germline origin. Copy number analyses indicated a loss of the wildtype *TP53* allele in the tumor, consistent with a second hit, while it was the variant *CHEK2* allele that was lost in the tumor. Our data indicate that the germline *TP53* PV acts as a driver of tumorigenesis in the reported case and support a complex interaction between the germline *TP53* and *CHEK2* PVs. This case highlights the dynamic interplays of genetic alterations in tumorigenesis and emphasizes the need for continued investigation into the complex interactions between *TP53* and *CHEK2* PVs and into the association of undifferentiated embryonal sarcoma of the liver and Li–Fraumeni syndrome.

## 1. Introduction

Genetic predisposition is increasingly recognized in children and adolescents with cancer [1,2]. To identify patients with an increased probability of an underlying cancer predisposition syndrome (CPS), various questionnaires and mobile apps have been developed. These tools incorporate data on clinical features, treatment toxicity, cancer subtype, somatic mutational spectrum, and family (cancer) history [3,4,5].

Sarcomas can arise in the setting of CPSs. Pathogenic variants (PVs) in cancer predisposing genes (CPGs) have been reported in 7–33% of children, adolescents, and adults with various types of soft-tissue sarcomas (reviewed in [6]). The most frequently mutated gene in patients with sarcoma is the cell cycle/cell death regulator tumor suppressor *TP53*, followed by the Ras/MAPK pathway inhibitor neurofibromin 1 (*NF1*), and the DNA repair regulators breast cancer associated 1 and 2 (*BRCA1* and *BRCA2*).

Germline variants in *TP53* are associated with Li–Fraumeni syndrome (LFS), characterized by a high and early-onset cancer risk. The tumor spectrum is broad and includes soft-tissue sarcomas and bone tumors, brain tumors, hematologic malignancies, breast cancer, and adrenocortical carcinoma, among others.

PVs in *BRCA1/2* have been associated with adult-onset cancer, most notably hereditary breast and ovarian cancer (HBOC) [7]. BRCA1 and BRCA2 are essential regulators of the homologous recombination repair (HRR) pathway and their inactivation results in impaired DNA-double strand-break repair and HRR deficiency (HRD). Additional genes participating in the HRR pathway, such as *ATM*, *ATR*, *CHEK1*, and *CHEK2*, the *BRCA2* loading factor *PALB2*, the recombinase *RAD51*, and the DNA–interstrand crosslink repair regulators of the *FANC* protein family contribute to the HRR pathway as well [8]. Recently, germline PVs in the HRR genes have recurrently been identified in children and adolescents diagnosed with cancer [1,9,10,11], although the clinical significance of these PVs in the context of pediatric cancer is only beginning to emerge [12]. 

A notable example of such pediatric cancers is undifferentiated embryonal sarcoma of the liver, a rare hepatic tumor of mesenchymal origin mainly occurring in children [13]. Therapeutic modalities for this entity encompass a multimodal strategy including chemotherapy, surgery, radiotherapy, and liver transplantation as appropriate. Data on the potential association between this tumor entity and CPS are, however, limited [14,15].

This case report uses the example of an undifferentiated embryonal sarcoma to illustrate the complex functional interplays of PVs in different genes against the background of Li–Fraumeni syndrome. It also aims to raise awareness of the need for comprehensive genetic profiling and a complete understanding of the genetic landscape of pediatric cancers.

## 2. Results

### 2.1. Case History

We report the case of an 18-year-old male patient with recurrent undifferentiated embryonal sarcoma of the liver. The patient initially presented at the age of 9.1 years with swelling of the left upper abdomen, vomiting, and recurrent abdominal pain. Following diagnosis (Figure 1), the patient was treated according to the Cooperative Soft Tissue Sarcoma (CWS) guidance. After three courses of neoadjuvant chemotherapy (I^2^VA: ifosfamide 3000 mg/m^2^ day 1 and 2 for each course, vincristine 1.5 mg/m^2^ day 1, 8, and 15 for courses 1 and 2, day 1 for all other courses, actinomycin D 1.5 mg/m^2^ day 1 for all courses, except courses 7 and 8), a left hemihepatectomy including complete tumor resection was performed, followed by six courses of adjuvant chemotherapy. Radiotherapy of the tumor bed was administered with up to 41.4 Gy.

Both ultrasound and MRI-based scans were performed as part of the follow-up care and in the further course of the disease history as preventive examinations. Since the onset of the disease, the patient has undergone a total of 41 ultrasound scans, all of which covered the abdomen. At least two ultrasound scans were performed per year, usually every two to three months. Up to the age of 15, 11 MRI scans were performed at approximately 6-month intervals. In the further course, annual whole-body MRIs were conducted. 

Two years later, at the age of 11, a suspicious lesion in the liver in segments 5/6 was detected by a routine ultrasound scan. Sonographic-controlled liver biopsy and subsequent partial atypical liver resection were performed. Histopathological evaluation demonstrated a regenerative node.

Two years after that, at the age of 13, another new lesion in the liver in segment 7 was identified during a routine ultrasound scan. Due to its increase in size, the lesion was subsequently excised through partial atypical liver resection. Histopathological evaluation was challenging, with differential diagnoses including mesenchymal hamartoma, intrasinsusoidal spread of the known sarcoma, and other sarcoma types. Ultimately, after a thorough review, three pathologists reached a consensus, diagnosing the lesion as a mesenchymal hamartoma with slight pleomorphism.

Another two years later, at the age of 15, additional lesions in segments 6/7 were detected by a routine ultrasound scan. Biopsy revealed a myxoid lesion with a proliferation rate of 5–15%, corresponding to the intrasinusoidal spread of the previously diagnosed undifferentiated embryonal sarcoma (Figure 2). Immunohistochemistry revealed positivity for CD56 but a lack of hepar 1 and glypican expression, confirming that the lesion was most likely not a neoplasm of primary liver origin. Systemic therapy was initiated according to the CWS guidance VACA-2 regimen, including vincristine 1.5 mg/m^2^ day 1 each course, actinomycin D (not administered), cyclophosphamide 1.200 mg/m^2^ day 1 each course, and doxorubicin 20 mg/m^2^ day 1–3 course 1 and 2. Following two courses of systemic chemotherapy, orthotopic liver transplantation was performed 4 months following diagnosis of relapse. Since then, the patient has been treated with ciclosporin and everolimus for immunosuppression. 

Three years later, at the age of 18, a routine whole-body magnetic resonance imaging (MRI) revealed several ambiguous osseous lesions in multiple vertebrae. A follow-up MRI three months later indicated disease progression, raising the suspicion of distant metastases of the undifferentiated embryonal sarcoma (Figure 3). To further characterize these lesions, an ^18^F-fluorodeoxyglucose (FDG) positron emission tomography (PET)/computed tomography (CT) was performed, demonstrating a moderate uptake of the lesions. A subsequent CT-guided biopsy confirmed a second (metastatic) relapse, showing small focally accentuated spindle cell infiltrates of an unclassified tumor compatible with the pre-diagnosed undifferentiated sarcoma. In immunohistochemistry, the tumor tissue showed a complete loss of p53 and regularly maintained Rb1 expression. Additionally, the tumor was negative for most immunohistochemical markers (CKAE1/AE3, p63, CD23, CD21, MUC4, NUT, CDK4, Pan-TRK, STAT6, SS18, MDM2, Desmin, S100, ERG, TLE1, Fli) and showed only weak and most probably unspecific expression of CD99, CD56, and CD34.

Figure 4 gives a graphical overview of the clinical course of the patient. 

### 2.2. Comprehensive Genetic Tumor Analyses

To identify therapeutic targets, the comprehensive genetic analyses of tumor tissue from the first relapse were performed within the Pediatric Targeted Therapy 2.0 project [16]. DNA methylation analysis demonstrated no concordance with established methylation classes within the sarcoma classifier, where the highest score corresponded to ‘well/dedifferentiated liposarcoma’. Numerous chromosomal gains and losses were observed in the copy number variation analysis. Immunohistochemical analysis for p-Akt and p-S6 demonstrated an activation of the mTOR signaling pathway (H score ≥ 100), constituting a potential therapeutic drug target. PVs in *TP53* and *CHEK2* were identified in the tumor tissue, suggesting germline origin. 

To obtain further details about the genetic landscape of the lesion, an additional next-generation sequencing analysis (NGS) of the tumor tissue from the first relapse was initiated. NGS with the TruSight Oncology 500 HRD assay (Illumina^®^) confirmed the previously identified PVs in *TP53* and *CHEK2* from the initial genetic analysis. Specifically, the *TP53* PV (NM_000546.5):c.532del p.(His178Thrfs*69) in exon 5 of 11 exons) was observed with a variant allele frequency (VAF) of 78%, and the *CHEK2* PV (NM_007194.4):c.85C>T p.(Gln29*) in exon 2 of 15 exons) was detected with a VAF of 25% in the tumor tissue. Copy number analyses, adjusted for tumor purity, indicated an absolute copy number of three for both the *TP53* and the *CHEK2* genes in the tumor specimen, in addition to further copy number alterations. No additional driver mutations classified as oncogenic or likely oncogenic were detected. The tumor mutational burden was determined to be low (8.8 mutations per megabase), and the microsatellite status was stable (0.96%). The genomic instability score (GIS) [17] was 36, which appears to be elevated; however, established cutoffs for this specific entity are not available. All NGS findings are summarized in the variant call file, which is available in the Appendix A.

### 2.3. Molecular Genetic Testing of TP53 and CHEK2 Genes

Germline genetic testing using the peripheral blood of the patient confirmed the diagnosis of Li–Fraumeni syndrome and HBOC, with both variants [*TP53* (NM_000546.5):c.532del p.(His178Thrfs*69), heterozygous, and *CHEK2* (NM_007194.4):c.85C>T p.(Gln29*), heterozygous] being of germline origin. Since then, the patient was followed up according to LFS surveillance recommendations [18]. Family history has been unsuspicious on both parental sides across three generations. Both parents have refused to undergo genetic testing so far.

## 3. Discussion

This report presents a unique case of a child with a very rare liver cancer, carrying double heterozygous germline PVs in the *TP53* and *CHEK2* genes. The simultaneous presence of germline PVs in *CHEK2* alongside other predominantly high-penetrance CPGs has been documented previously [19]. In HBOC patients, double or—more general—multiple heterozygosity commonly involves PVs in *BRCA1* or *BRCA2*, with less frequent instances involving PVs in *ATM*, *CHEK2*, and other moderate-risk CPGs [20,21]. In contrast, double heterozygosity among *TP53* carriers is exceptionally rare and involves, amongst others, co-occurring PVs in genes such as *PALB2* and *ATM* [22]. To the best of our knowledge, this is the first report of co-occurring PVs in *TP53* and *CHEK2* in a pediatric cancer patient. Double heterozygous PVs in both genes are extremely rare in adults but have previously been reported, e. g., in breast cancer patients [23].

The *TP53* gene encodes p53, which plays a key role in regulating the cellular response to DNA damage by regulating DNA repair, apoptosis, and growth arrest [24]. Upon genome damage, p53 is activated through post-translational modifications, including phosphorylation by DNA damage-activated protein kinases including ATM, ATR, CHEK1, and CHEK2 at specific sites. Phosphorylation at Serine 20 by the CHEK2 in response to DNA damage stabilizes p53 by inhibiting its inactivation with the negative regulator MDM2. While some experimental studies have suggested a critical role of CHEK2 in activating the p53 apoptotic response to genotoxic stress [25,26], other studies have suggested that CHEK2 may be dispensable for p53 activation concerning apoptosis and growth arrest [27]. Since p53 can also suppress carcinogenesis by regulating DNA repair, this may provide an additional explanation for CHEK2-mediated effects on p53.

Both *TP53* and *CHEK2* are classified as tumor suppressor genes. Given that the *TP53* PV in this patient is a frameshift variant, it is not anticipated to exhibit a dominant-negative effect [28]. According to the two-hit hypothesis proposed by Knudson, tumorigenesis requires the inactivation of both alleles of a tumor suppressor gene [29]. Loss of heterozygosity (LOH), which results from the deletion of one allele, represents a well-established mechanism for this second hit. LOH events can be followed by duplications of the retained allele, leading to either copy-neutral loss LOH or, when overcompensated, to copy-gain LOH [30]. The tumor analyses presented in the patient reported herein support a copy-gain LOH mechanism, suggesting that the tumor cells may harbor three mutated *TP53* alleles, while the mutated *CHEK2* allele is lost (Figure 5). The absence of functional *TP53* alleles and the resultant loss of *TP53* function are thus posited as significant drivers of cancer development in this patient.

The loss of the variant *CHEK2* allele, despite the presence of a germline PV, is striking. This observation aligns with a model suggesting that the co-occurrence of a loss of *CHEK2* function with a loss of *TP53* may establish a synthetic lethal interaction. Synthetic lethality refers to a genetic interaction concept wherein the simultaneous loss of function of two genes leads to cell death, whereas the loss of either gene alone permits cell viability [31,32]. As a consequence of synthetic lethality, only cells that have lost either *TP53* or *CHEK2* (or retain functional activity of both genes) may be observed. Within this framework, *CHEK2* mutations have been proposed to exhibit synthetic lethality in conjugation with *TP53* mutations [33].

The interplay between *TP53* and *CHEK2* in the patient presented herein underscores the necessity for a nuanced understanding of the genetic landscape in pediatric cancers [12]. The presence of multiple genetic alterations can lead to complex interactions that significantly influence tumor behavior. Therefore, further investigation into the functional consequences of these PVs, as well as their interactions, is essential to elucidate their roles in tumor development and progression. This case not only links undifferentiated embryonal sarcoma of the liver with the Li–Fraumeni syndrome but also highlights the importance of comprehensive genetic analysis involving tumor and germline analyses in understanding pediatric cancers effectively.

## 4. Methods

Molecular genetic tests were performed on genomic DNA samples isolated from the peripheral blood leukocytes using the QIAamp DNA Blood Mini Kit (Qiagen, Hilden, Germany). CNV analysis was also performed based on coverage using NGS data. 

Tumor DNA was extracted from the FFPE samples using the Maxwell^®^ CSC DNA FFPE Kit (Promega; Madison, WI, USA), and nucleic acids were quantified using the Quantus System (Promega). Library preparation was performed using the hybrid capture-based TruSight Oncology 500 HRD Library Preparation Kit (Illumina; San Diego, CA, USA) according to the manufacturer’s protocol. Finally, the libraries were pooled, denatured, and diluted to the appropriate loading concentration. TSO500 libraries were sequenced on a NextSeq™ 550Dx system (Illumina). For a secondary analysis, raw data were analyzed using DRAGEN TruSight Oncology 500 Analysis Software 2.5.2 on a local DRAGEN server (Illumina).

Sequence variants were described using HGVS nomenclature [34]. Variant classification was based on the guidelines of the American College of Medical Genetics and Genomics (ACMG) [35].

## 5. Conclusions

This case report presents the unique occurrence of double heterozygous pathogenic variants in *TP53* and *CHEK2* in a pediatric patient with undifferentiated embryonal sarcoma of the liver. Our findings underscore the potential role of *TP53* as a key driver in tumorigenesis, with the additional loss of *CHEK2* possibly facilitating tumor development through a synthetic lethal mechanism. This case contributes valuable insights into the complex interplays of genetic factors in pediatric liver sarcoma, advocating for further research into the interactions between the *TP53* and *CHEK2* mutations. It highlights the necessity of comprehensive genetic screening in pediatric cancer patients, which can lead to improved risk stratification, personalized surveillance, and targeted therapeutic strategies.

## Figures and Tables

**Figure 1 ijms-25-11489-f001:**
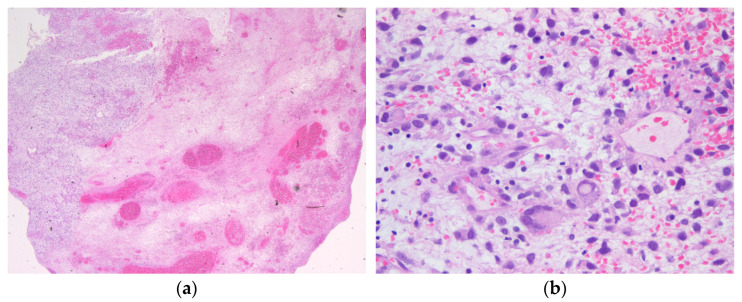
Hematoxylin and eosin staining of the biopsy tissue showing a myxoid and necrotic tumor with pleomorphic tumor cells, giant cells, and partial rhabdoid features that were classified as undifferentiated embryonal sarcoma of the liver (**a**) 12.5× magnification; (**b**) 400× magnification. Immunohistochemistry revealed positivity for CD56, but negativity for SMA, desmin, pancytokeratin, CD99, myogenin, MyoD1, MDM2, S100, EMA, WT1, and CD34 in the tumor cells. The Ki67 proliferation index was measured up to 40% in the sample.

**Figure 2 ijms-25-11489-f002:**
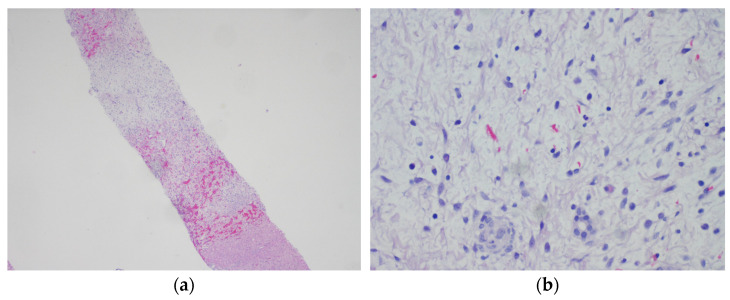
Hematoxylin and eosin staining of the recurrent undifferentiated embryonal sarcoma of the liver at the age of 15 (**a**) 12.5× magnification; (**b**) 200× magnification.

**Figure 3 ijms-25-11489-f003:**
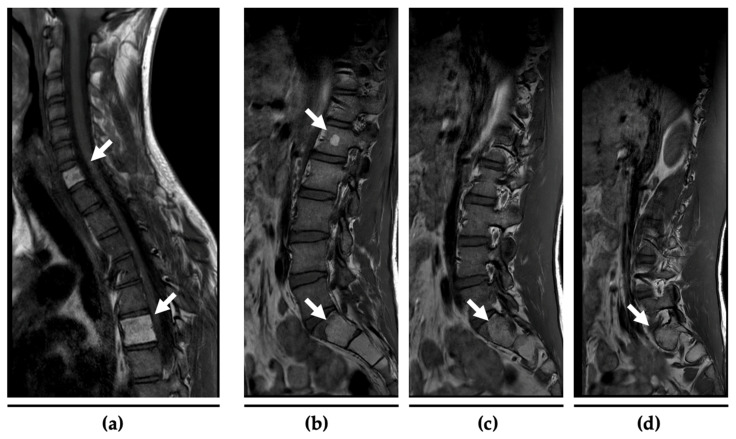
MRI scans of the cervical/thoracic spine (**a**) and the lumbar spine/sacral bone (**b**–**d**) at the age of 18. The white arrows mark the lipomatous lesions of the vertebral bodies C7 and Th6 as well as the rounded lipomatous focus in L1 and the mass in S1–3 from which the biopsy was taken.

**Figure 4 ijms-25-11489-f004:**
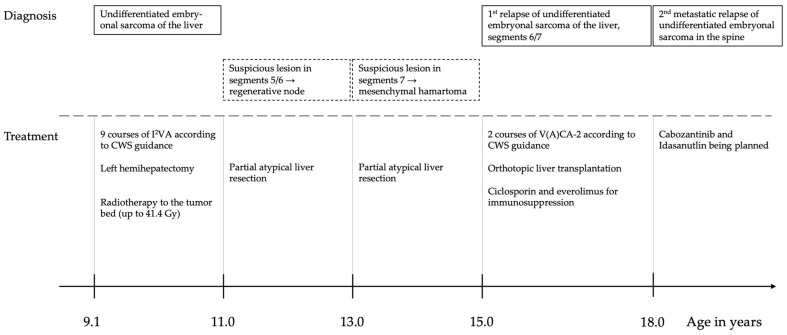
Graphical overview on the clinical course of the patient.

**Figure 5 ijms-25-11489-f005:**
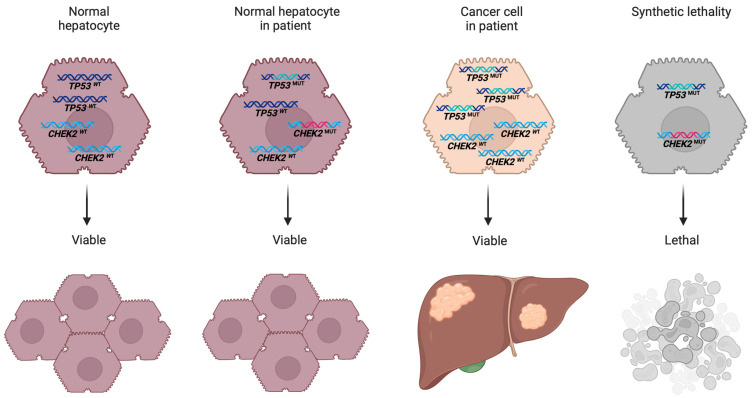
Copy number alteration and synthetic lethality interaction in tumor tissue: Loss of variant *CHEK2* allele and gain of mutant *TP53* allele in pediatric undifferentiated embryonal sarcoma.

## Data Availability

To protect patient privacy, the patient’s data are not available for public access, but all data from this manuscript are available from the corresponding author upon reasonable request.

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
