# Peer review of "Double Heterozygous Pathogenic Variants in TP53 and CHEK2 in Boy with Undifferentiated Embryonal Sarcoma of the Liver"

_ijms, 2024, doi:10.3390/ijms252111489_

Round 1
Reviewer 1 Report
Comments and Suggestions for Authors
The submitted manuscript presents the case report on the double heterozygous pathogenic mutations in TP53 and CHEK2 in a male patient with undifferentiated embryonal sarcoma of the liver.
The manuscript is well written, concise, informative. The presented case is rare and important at the same time. There are some editorial mistakes, i.e. I don’t know why the references are highlighted with yellow color. However, taking into account that the type of the article is “case report”, the manuscript can be accepted after minor revision.
Minor revisions:
It should be stated how often were the ultrasound and MRI scans performed, starting from age 9.1
Line 66, at this moment the authors should briefly state the aim of the study
Line 76, please list the APIs used for adjuvant chemotherapy
Lines 87 and 94 and 105, “years” can be removed
Line 93, it should be “as a”
Line 157, what about the adult population?
Line 203, here a conclusions sentence/section should be added
References: the comment of the editor regarding the references style was not addressed, so far, but I guess it will be updated
Reviewer 2 Report
Comments and Suggestions for Authors
Manuscript entitled "Double heterozygous pathogenic variants in TP53 and CHEK2 in a boy with undifferentiated embryonal sarcoma of the liver"
Major issues:
1. The authors should provide a table summarizing the NGS findings in both liver tumor and germline material. The SNV/Indel, CNV, Fusion should be provided. The authors are encouraged to provide the vcf file as the supplementary files.
2. The immunohistochemistry profile of the tumors should be provided.
3. The radiographic image of this case should be provided.
4. A figure summarizing the history and therapeutic protocol should be provided.
Round 2
Reviewer 2 Report
Comments and Suggestions for Authors
The revision is acceptable.